# A Comparative Evaluation of the Structural and Dynamic Properties of Insect Odorant Binding Proteins

**DOI:** 10.3390/biom12020282

**Published:** 2022-02-09

**Authors:** George Tzotzos

**Affiliations:** Ferrogasse 27, 1180 Vienna, Austria; gtzotzos@me.com

**Keywords:** insect OBPs, structural analysis, covariance similarity analysis, elastic network models

## Abstract

Insects devote a major part of their metabolic resources to the production of odorant binding proteins (OBPs). Although initially, these proteins were implicated in the solubilisation, binding and transport of semiochemicals to olfactory receptors, it is now recognised that they may play diverse, as yet uncharacterised, roles in insect physiology. The structures of these OBPs, the majority of which are known as “classical” OBPs, have shed some light on their potential functional roles. However, the dynamic properties of these proteins have received little attention despite their functional importance. Structural dynamics are encoded in the native protein fold and enable the adaptation of proteins to substrate binding. This paper provides a comparative review of the structural and dynamic properties of OBPs, making use of sequence/structure analysis, statistical and theoretical physics-based methods. It provides a new layer of information and additional methodological tools useful in unravelling the relationship between structure, dynamics and function of insect OBPs. The dynamic properties of OBPs, studied by means of elastic network models, reflect the similarities/dissimilarities observed in their respective structures and provides insights regarding protein motions that may have important implications for ligand recognition and binding. Furthermore, it was shown that the OBPs studied in this paper share conserved structural ‘core’ that may be of evolutionary and functional importance.

## 1. Introduction

Insects are the dominant animal species on earth. Their remarkable evolutionary success is largely attributed to their highly intricate olfactory systems. Early olfactory processing takes place at the peripheral nervous system, and involves solubilisation, and transport of semiochemicals (pheromones, and odorant or sapid molecules) to olfactory receptors (ORs) [1,2]. ORs are located on the dendrites of sensory neurons and bathed in an aqueous medium known as the sensillar lymph. They mediate the detection of chemical cues. Further processing of olfactory signals occurs in the central nervous system resulting in odour perception and thence adapted behaviour. Other protein components of the peripheral olfactory system include odorant binding proteins (OBPs), odorant degrading enzymes (ODEs) involved in the inactivation of excess odorants following OR activation, sensory neuron membrane proteins (SNMPs) functioning in tandem with ORs [3], and ionotropic receptors (IRs), which are thought to be an evolutionarily different mechanism of chemoreception [4]. OBPs, which are the subject of this review, are implicated in the solubilisation, binding and transport of semiochemicals to ORs.

The first insect OBP was isolated from the giant moth *Antheraea polyphemus* in the early 1980s [5]. Since then, the number of insect proteins classified as OBPs has grown enormously. The PFAM database [6] lists over 3000 sequences from 104 species classified as pheromone-binding and general odorant-binding proteins (Pfam ID: PF01395). The number of OBPs per insect species varies widely. For example, 69 OBPs have been associated with *A. gambiae*, whereas 52 have been associated with *D. melanogaster*, 21 with *A. mellifera* and 13 with *B. mori* [7,8].

Insects dedicate an enormous part of their metabolic resources to synthesise OBPs in their antennae, where their concentration may be as high as 10 mM [9]. RNAseq analysis has shown that the levels of RNA for the most abundantly produced OBPs in the olfactory segment of the Drosophila antenna are three orders of magnitude higher than those of ORs [7]. For many years, the prevailing hypothesis regarding the role of OBPs in insect olfaction associated them with the solubilisation, binding and transport of semiochemicals to ORs. Strong support for this hypothesis came from studies involving pheromones (PBPs) and general odorant binding proteins (GOBPs) in Lepidopteran species demonstrating high specificity of these proteins for volatile compounds [10,11]. Added support for this hypothesis came from studies suggesting a possible release mechanism of bound odorants brought about by conformational changes in OBPs induced by the negatively charged environment at the dendritic membranes.

However, the above hypothesis has been challenged by research showing that OBPs:display broad specificity of binding, and ability to bind more than one ligand simultaneously [12,13].are expressed not only in the insect olfactory organs but also in other body parts [14]. For example, it was shown in proteomic studies that of the 66 genes encoding OBPs in *Anopheles gambiae* only 18 are detectable in olfactory tissues [15]. These results raise questions regarding the actual number of OBPs that may have a purely olfactory function.

In the light of the above, it is now thought that OBPs may play diverse, as yet uncharacterized, roles in insect physiology. Thus, in addition to the odorant transport hypothesis, OBPs may be involved in the protection of odorants from degradative enzymes, odorant clearance from the sensillum lymph after OR activation, or buffering against sudden changes in odour levels after the termination of an odorant pulse [7]. For OBPs expressed in non-olfactory tissues, it has been proposed that they may play a role in the modulation of mating behaviour, haematopoiesis, humidity detection and attraction or aversion of gustatory cues [8].

The rapidly increasing numbers of OBP structures, solved by X-ray crystallography and NMR, serve as a solid foundation to draw functional inferences. The number of structures deposited at the Protein Data Bank (PDB) has grown from 4 in 2004 [16] to 25 derived from 17 species belonging to 7 insect orders. These OBPs, known as “classical” OBPs, are globular proteins consisting of alpha-helices of different topologies bearing a characteristic signature of three conserved disulfide bonds. Four of the cysteines are equispaced with three amino acids between the second and the third cysteines (C_2_, C_3_), and eight amino acids between the fifth and the sixth cysteines (C_5_, C_6_). The disulfide bonds form the pattern C_1_–C_3_, C_2_–C_5_, C_4_–C_6_. “Classical” OBPs vary in the size and topology of their respective helices, and belong in the same PFAM family (PF01395) and have same fold in SCOPe (Fold a.39: EF Hand-like) [17]. In addition to the “classical” OBPs, other subgroups differ in the number of disulfide bonds: the so-called C-plus OBPs with >3 disulphide bonds, and the so-called C-minus OBPs with <3 disulfide bonds.

OBP structures have been described in detail by numerous papers as well as in several reviews [2,18,19,20]. Yet, X-ray structural models fail to capture structural fluctuations due to natural thermal motions, which are induced by equilibrium fluctuations and non-equilibrium effects [21]. Equilibrium fluctuations determine protein action, and as such, the dynamics of protein motion are critical in facilitating protein interactions, or even driving them. The 3D architecture of a protein, or fold, characterise its conformational space under physiological conditions, and this, in turn, circumscribes the spectrum of motions that are intrinsically accessible to the interacting partners such as odorants or proteins. By implication, evolutionary pressure may have selected specific motions that enable proteins to perform their biological functions. This process is encapsulated in the evolutionary paradigm “structure-encodes-dynamics-encodes-function” [22].

The intrinsic motions of biomolecules are usually described in terms of the spectrum of the available vibrational frequencies or normal modes (see Appendix E). The low-frequency motions are highly collective, meaning that they involve cooperative movements of large portions of the structure, such as domain motions or structural rearrangements. These motions, frequently referred to as global or intrinsic motions, provide a useful measure of the relative rigidity in proteins. A small set of the lowest frequency vibrational modes suffices to quantify the large-scale protein motions [23]. The higher frequency modes are less collective and correspond to local changes in conformation and dynamics restricted to a few atoms. They are localised in the interior or surface of the protein and are strongly implicated in signal transmission or other internal processes [24]. Normal mode analysis (NMA) provides a powerful tool for the study of protein structure and dynamics. However, NMA is very expensive computationally and for this reason simplified coarse-grained models, such as the elastic network models (ENMs) have been developed for efficient normal mode computations [25] (see SI NMA). ENMs have been used extensively to study protein dynamics [23,26].

The aim of this review is to provide a comparative analysis of the similarities and dissimilarities of OBPs in terms of their structural and dynamic properties. The focus is not on individual OBPs but rather on this group of proteins as a whole.

## 2. Materials and Methods

The OBPs used in this study comprise exclusively the “classical” OBPs (Table 1). In order to ensure the quality of structural comparisons, classical OBPs with truncated structure (PDB IDs: 3l4a) or of low resolution (PDB IDs: 3l4a, 6vq5) or solved by NMR (PDB IDs: 2jpo, 1tuj, 6um9) were excluded. For the same reason, C-plus and C-minus OBPs were likewise excluded. The 20 OBPs in the dataset (Table 1) were compared in terms of sequence and structure similarity. They were then subjected to principal component analysis (PC) to identify patterns of structural variance, and finally to normal mode (NM) analysis using elastic network models (ENM) to determine their dynamic properties [27].

The Bio3d v2.4-2 suite of programs was used for sequence, structural, principal component and normal mode analyses [28,29]. The nma function of Bio3d was used to for normal mode calculations employing an elastic network model (ENM) using the ‘calpha’ force field. This force field employs a spring force constant differentiating nearest-neighbour pairs of atoms along the backbone from pairs in spatial proximity. The resulting mode vectors were scaled by the thermal fluctuation amplitudes.

UCSF Chimera (Structure Measurements panel, Axes/Planes/Centroids section) was used for structure measurements described in Section 3.2 [30]. The distance and angles measurements were made based on sets of selected helical residues. The protein structure visualisation was performed with PyMOL v2.5.2 (Schrodinger pymol-open-source).

The EvoCouplings method was used for to find the residues in each OBP with the highest number of coupled interactions to their neighbours. The method is based on the direct information (DI) to compute a set of direct residue couplings that best explains all pair correlations observed in the multiple sequence alignment [31]. The EvoCouplings implementation of the methods is achieved through an open-source Python package for coevolutionary analysis [32,33].

As a prerequisite for the structural comparison of OBPs, we constructed a multiple structural alignment (MSA). Structural alignments have been found to perform better for the deduction of the structural and dynamic relatedness of the OBPs than the corresponding sequence-based ones, particularly in cases of low sequence identity [34]. Furthermore, the choice of the structural alignment method is critical because different alignment algorithms make different assumptions regarding the length of the consensus sequence, and the pairwise root-mean-square deviations (RMSD) within the set of aligned structures [35]. A local installation of Mustang v.3.2.3, interoperable with Bio3d, was used to construct the MSA [36]. The length of the sequences in the dataset ranges between 115 to 142 amino acids (average length 124 amino acids). The resulting MSA consists of 198 columns, 93 of which have 100% occupancy, i.e., columns containing no gaps (Appendix B Figure A1). An MSA in which columns with gaps have been removed is shown in Appendix B Figure A2.

## 3. Results

### 3.1. Sequence Analysis

A percent identity value, in the form of a numeric score, was determined for each pair of aligned sequences. The identity score, which was normalised to values between 0 and 1, measures the number of identical residues in relation to the length of the alignment (Figure 1). The summary statistics of the calculated pairwise sequence identities are given in Table 2. Typically, the average sequence identity over all pairs in the dataset is around ~24%.

For most OBPs, the pairwise sequence identities are low, placing them in so-called “twilight-zone” where sequence identity falls between 10 and 30% (Figure 1). The “twilight-zone” is an operational term setting the boundaries of confidence for detecting evolutionary relatedness between proteins [37,38]. Despite the low sequence identities, the OBP structures have retained the same fold albeit with differences in the geometries of the packed secondary structure elements.

### 3.2. Structure Analysis

To compare the similarity of the OBP structures, we used the structure coordinates of the Cα-atoms and quantified it using the optimal RMSD over the sites that aligned in the MSA. RMSD is a similarity measure varying between 0 and ∞ [39]. The calculation was made from the 93 equivalent positions in the MSA. The summary statistics of the calculated differences in RMSD are given in Table 3, and the heatmap and related dendrogram derived from the pair-wise similarity matrix of RMSD values (Figure 2) displays the structural relatedness of the OBPs.

The dendrogram shows two main clades that split at 3 Å. One of these clades consists of 11 OBPs from 4 different insect orders, namely Diptera [AaegOBP1(3k1e), AgamOBP1 (3n7h), CquiOPB1 (3ogn), AgamOBP4 (3q8i), AgamOBP20 (3v2l), DmelOBP76a (2gte), CcapOBP22 (6hhe), Hymenoptera [AmelASP1 (3bjh), AmelAsp5 (3r72)], Orthoptera [LmigOBP1 (4pt1)], and Neuroptera [CpalOBP4 (6jpm)]. Three mosquito proteins (AgamOBP1, AaegOBP1, CquiOBP1) are homologous (sequence identity > 80%), displaying the greatest structural similarity (RMSD 0.3–0.36 Å). The range of RMSD differences amongst the rest of the OBPs in this clade is between 1.1 Å and 3.0 Å.

The second clade comprises OBPs that exhibit the greatest structural differences. They form the leaves of 4 distinct branches. One of the branches includes the aphid OBPs [MvicOBP3 (4z39), NribOBP3 (4z45)]. A second branch, the moth OBPs [BmorPBP1 (1dqe), BmorGOBP2 (2wc5), AtraPBP1 (4inw)]. The third branch, LmadPBP1 (1org) and DmelOBP28a (6qq4), and the fourth branch, AaegOBP22 (6oii), PregOBP56a (5dic). All of these proteins are expressed in olfactory tissues with the exception of the last two. PregOBP56a is specifically expressed in a cluster of cells on the oral disk of the insect. PregOBP56a solubilises fatty acids from meat meal during feeding and delivers them to the midgut where it may help the process of reproduction [40]. AaegOBP22 is expressed in multiple tissues including the antenna, and the male reproductive glands. Expressed in different antennal tissues, AaegOBP22 may modulate chemosensory responses, but as a secreted protein in the salivary gland is likely to be injected into the host during a blood meal [41].

The structural alignment of OBPs in the dataset involves a procedure in which (i) a score was calculated from pairwise residue-residue correspondences, followed by pairwise structural alignments; (ii) recalculation of scores in the context of multiple structures, and (iii) merging the series of pairwise alignments along a guide tree [36]. Visualisation of the superimposed structures shows helical regions where the RMSD difference of the backbone carbon atoms is the least (Figure 3, Panel a). To identify residues with the least RMSD difference, the aligned structures were subjected to an iterated superposition procedure, each round of which identified and eliminated residues displaying the largest positional differences. The subset of retained residues implies an evolutionarily conserved structural “core” consisting of 17 residues. In this case, 14 of these including OBP conserved cysteines C5 and C6, are found in the carboxy-terminal helix; the remaining 3 are found in the helix which contains C_2_. The pair-wise RMSD difference of the “core” residues is <1 Å. The angle between the two helices is in the range of ∼78–88° and the distance between them is in the range of 6.8–8.1 Å (Figure 3, Panel b). The aphid OBPs (4z39, 4z45), and PregOBP56a (5dic) show the greatest deviations from this geometry (Appendix C, Table A1).

### 3.3. Principal Component Analysis (PCA)

PCA was used to gain insights that best explain the underlying patterns of variance in the ensemble of aligned structures. When performed on an ensemble of interpolated X-ray structures, PCA captures concerted atomic displacements highlighting major structural differences between the structures. The basic premise of PCA is that it is possible to eliminate highly correlated variables in the data set without losing essential information (Appendix F).

The PCA calculations were based on equivalent residues in the MSA. Approximately 82% of the total mean-square displacement (variance) of Cα atom positional fluctuations is captured by six principal components, namely the axes of maximal variance of the distribution of structures (Figure 4). The first three principal components account for 71% of the structural variance. The eigenvalues capture the percentage of the total variance (i.e., the total mean square displacement) of atom positional fluctuations.

The influence of the individual OBPs on the structural variation and correlations within the ensemble of structures can be captured by means of score plots (see Appendix F). The score plot of the first two PCs, accounting to ~60% of the structural variance, maps the interrelation of the OBP structures (Figure 5a). In any given cluster depicted in the plot, the deviation of the equivalent Cα atom positions is the least. Comparing the PCA clustering to the corresponding one obtained from the RMSD similarity matrix (Figure 5b), it can be safely concluded that the score plot reflects the relatedness of OBPs as observed from the pairwise structural comparisons.

The contribution of the individual residues, also known as loadings, to the structural variance covered by the first three PCs is presented in Figure 6. The figure shows that the least of the structural variance occurs in the 3rd and 6th helical regions where the residues of the structural “core” are located.

### 3.4. Normal Mode Analysis (NMA)

The intrinsic dynamics and flexibility of the OBPs were determined by the use of normal mode analysis based on the Elastic Network Models described in Section 2. The structural flexibility of proteins can be decomposed to high-frequency localised fluctuations of the individual amino acids, and to low-frequency motions of large rigid domains.

#### 3.4.1. Covariance Similarity Analysis

To compare the similarity of the low-frequency motions of the OBPs, the 10 lowest modes for all pairs of structures were calculated. Two points are worth noting. First, there is no single well-accepted measure to describe dynamics similarity, and consequently linking the dynamics space to the structure space remains a challenge [39]. Second, dynamical similarity scores are extremely sensitive to MSA, particularly in cases where the sequence identities are very low. The root-mean square inner product (RMSIP was used as a similarity metric [39]. RMSIP is sensitive to the number of modes used. Finer separations are discernible with smaller number of modes. When the number of modes included in the calculation approaches the full number, the dynamic distance order approaches the RMSD order. The RMSIP distance matrix was used to determine the relatedness of OBPs in terms of low-frequency modes.

The dynamics-based hierarchically-clustered heatmap or clustermap shows all pairwise comparisons for the aligned proteins (Figure 7). Comparison of a protein to itself along the diagonal have an RMSIP value equal to 1. Lower values indicate overlapping low-energy fluctuations of the aligned backbone carbon atoms. Fluctuations covering completely non-overlapping space have an RMSIP value equal to 0. The similarity clustering shown plot is in agreement with the structural classification of the OBPs as obtained from the pairwise RMSD measurements of the OBPs (*cf.* Figure 2).

#### 3.4.2. Comparative Fluctuation and Deformation Analysis

The amplitude and directionality of residue fluctuations provide a measure of the local flexibility in any given protein. Fluctuations are defined as the sum of each atom’s displacement along each mode, weighted by the reciprocal of the eigenvalues [42]. However, residue fluctuations are less informative regarding the amount of local flexibility of a given protein structure. Local deformation involves energy exchange arising from short-range atomic interactions. Atomic motions relative to neighbouring atoms can be calculated from the atomic energy contributions to deformation as a function of position. Thus, lower deformation energies may correspond to relatively rigid regions [24]. It is noted that deformation energies have no quantitative physical meaning, and therefore no units [43].

The fluctuation and deformation energy profiles of the conserved residues in the MSA of OBPs, averaged over all modes and normalised, are given in Figure 8.

The plot in Figure 8 shows that the largest amplitude fluctuations occur almost exclusively in non-helical regions. Residues with the highest deformation energies lie in the helical regions that contain C_2_, C_3_, C_5_ and C_6_ and are coincident with the conserved structural core. The high deformation energy of these residues is indicative of interactions with spatially proximal residues [24,42]. Favourable interactions between spatially proximal residues are a major constraint in the evolutionary variation of proteins belonging to the same fold. Evolutionary coupling analysis of the individual OBPs showed C_1,_ C_2,_ C_5_, and residues adjacent to them in the sequence, form ‘dense’ coupling interactions with residues that are in close 3D proximity (data available in Appendix A).

Moving beyond the structural and dynamic classification of OBPs, single structure analysis using GNM can provide additional inferences. Examples are given below.

#### 3.4.3. Single Analysis on Selected Examples

While the covariance matrix calculated from the low-frequency modes (see Figure 7) allows the classification of OBPs in terms of similarity of motions, it provides no information regarding their relative flexibilities. Protein motions entail the movement of relatively rigid structural domains, especially regions of secondary structure, with intervening regions of local flexibility [44]. During motion, these “dynamic” domains remain internally rigid in all conformations but move relative to each other. In OBPs such motions have been observed in studies of the *A. mellifera* antennal-specific protein, which at pH 7.0 is a domain-swapped dimer [45]. The relative flexibility of the OBPs in the dataset was determined by means of the Bio3D implementation of the Geometrically Stable Substructures (GeoStaS) algorithm [46]. Although the evaluation of residue cross-correlation coefficients is the standard method to detect atomic motions, GeoStaS has the advantage of detecting both rotations and translations, in contrast with residue cross-correlation analysis which detects translations only. The GeoStaS calculations based on the 5 lowest-frequency non-trivial modes showed considerable variation amongst the OBPs both in terms of the sizes of the “dynamic” domains, as well as the correlation of domain movements. This variation is exemplified in Figure 9 and additional material can be found in Appendix A.

Two additional examples are presented below. These involve AaegOBP22 (6oii) and DmelOBP28a (6qq4). The choice of these proteins was based on the availability of crystal structures both in free and liganded forms. It is noted that, in addition to these structures, few other structures in the dataset have been crystallised in ligand-free form, namely, BmorPBP1, BmorGOBP2, AgamOBP20 and AmelASP1. AaegOBP22 has been crystalised in open liganded form (PDB ID: 6oii) and closed ligand-free form (PDB ID: 6og0). Ligand binding has been associated with a conformational change at the carboxy-terminal end leading to the formation of an enlarged binding cavity [41]. The ligand-free structure is truncated missing 4 residues at the carboxy-terminus. The RMSD difference between the two structures is 1.35 Å. Subjecting the ligand-free structure to cross-correlation analysis shows significant anticorrelated motions involving residues of the 1st, 2nd and 6th helices (Figure 10, panel a). These can be visualised in the corresponding three-dimensional model of the protein (Figure 10, panel b). These motions may result in conformational changes that could accommodate ligand binding and the concomitant widening of the opening to the internal cavity of the protein.

In the case of DmelOBP28a, the protein in liganded and free (apo) forms revealed a large conformational reorganization induced by ligand binding. This conformational change involved the 1st, 4th and 5th helices [47]. The residue-residue cross-correlation matrix shown in Figure 11, panel a is consistent with the structural observations as it identifies residues helices 1, 4 and 5 involved in anti-correlated movements

The normalised fluctuations and deformation energies of Cα atoms of AaegOBP22 (PDB ID: 6oii) and DmelOBP28a (PDB ID: 6qq4, chain A), are included in Figure 7. Comparison of the fluctuation and deformation energy profiles against the corresponding conformers in ligand-free form, showed an increase in the magnitude of Cα atom mobilities in some of the interhelical regions (Appendix D, Figure 1 and Figure 2). Analysis of a greater number of models would be required to draw firm conclusions regarding the difference in fluctuations between ligand-free and ligand-bound forms.

## 4. Discussion

From the preceding analysis it is obvious that, in most cases, the sequence identities between pairs of OBPs fall below the twilight zone and, therefore, it is not possible to draw conclusions regarding the possibility of divergent evolutionary relatedness of these proteins [37]. In terms of structure, the pairwise RMSD calculations and PC analysis result in a clustering of OBPs into subgroups that reflects, in part, phylogenetic classifications. The foregoing analysis identifies a conserved structural ‘core’ that represents 15–20% of the total sequence length of the OBPs. The ‘core’ residues, which include C_2_, C_3_, C_5_ and C_6_, are located in the helical regions with the least structural variance (Figure 5 and Figure 6). In particular 14 of the 17 ‘core’ residues are found in the carboxy-terminal helix. The high deformation energies of the residues of these helical regions (Figure 8) are indicative of interactions with spatially proximal residues [24]. Further evidence of such interactions was derived from evolutionary coupling analysis. Evolutionary coupling interactions have a strong bearing on the overall architecture of the fold [31,48]. Generally, the similarity of 3D structure amongst the members of a particular fold is the result the constrained evolutionary boundaries ensuring conservation of function. This, in turn, implies the co-evolution of residues in spatial proximity in order to maintain energetically favourable interactions. Consequently, the high density of coupling interactions is indicative of ‘hotspots’ in the evolution of proteins and, as such, may be of functional importance [31]. From the preceding, it is not unreasonable to suggest that the residues adjacent to C_2_, C_3_, C_5_ and C_6_ may play an important role in ligand binding or recognition.

The covariance similarity analysis of the low-frequency motions of the OBPs (Figure 7) shows that their relatedness in terms of intrinsic dynamics mirrors that observed from the RMSD calculations and PC analysis (Figure 2 and Figure 5). Further insights were obtained from the analysis of the motions of the individual proteins. The analysis showed that in some OBPs the motions are highly collective and thus may be significant in protein-ligand interactions (see examples in Section 3.4.3) [44,49]. Structural flexibility ensures the predisposition of proteins to attain alternative conformations. The pool of accessible conformations allows proteins to achieve their biological function in the presence of ligand binding, which, in turn, depends on changes in external conditions shifting the equilibrium between conformational states. Thus, although the examples described in Section 3.4.3 point to the importance of the intrinsic motions in determining the functional dynamics of the OBPs, it would be unwise to make broad generalisations in the absence of knowledge of the physiological milieu in which these proteins function.

## 5. Conclusions

In this review, the “classical” insect OBPs were compared in terms of their structural features, and intrinsic dynamics. It was found that the structural differentiation of OBPs into subgroups is reflected in their low-frequency motions. The small dataset of available crystal structures was a limiting factor in the analysis. Without a sufficiently large dataset, it is not possible to make an in-depth evaluation of the relation between the conservation of structural dynamics to the evolution of sequence and structure in insect OBPs. Nevertheless, the study of OBPs, individually or collectively, by means of NMA using EN models can provide useful insights into the interplay between structure and dynamics at little computational cost. NMA using EN models has been applied to other heterogenous protein families of which the globin family is a case in point. The globin fold is an all-α fold made up of 6–7 helices that define the architecture of a well-defined haem pocket [50]. Although the divergent sequence evolution of the family induced changes in the relative disposition of helices, these had small effect on their packing. This is because the position of the haem group is essential for protein function [51]. In the case of OBPs, it is tempting to speculate that natural selection operated in the opposite direction by allowing promiscuous changes in architecture of the fold in order to permit fast adaptation to the diverse chemical ecologies of insect.

## Figures and Tables

**Figure 1 biomolecules-12-00282-f001:**
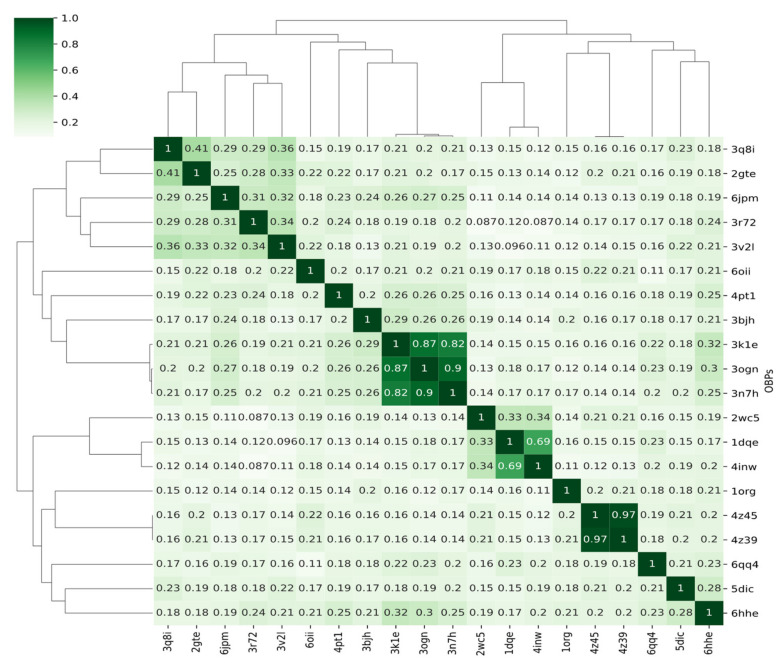
Pairwise sequence similarity of OBPs. Comparisons of a protein to itself are shown along the diagonal. The dendrogram shows the hierarchical clustering obtained from the complete linkage algorithm as implemented in the Python package Seaborn.

**Figure 2 biomolecules-12-00282-f002:**
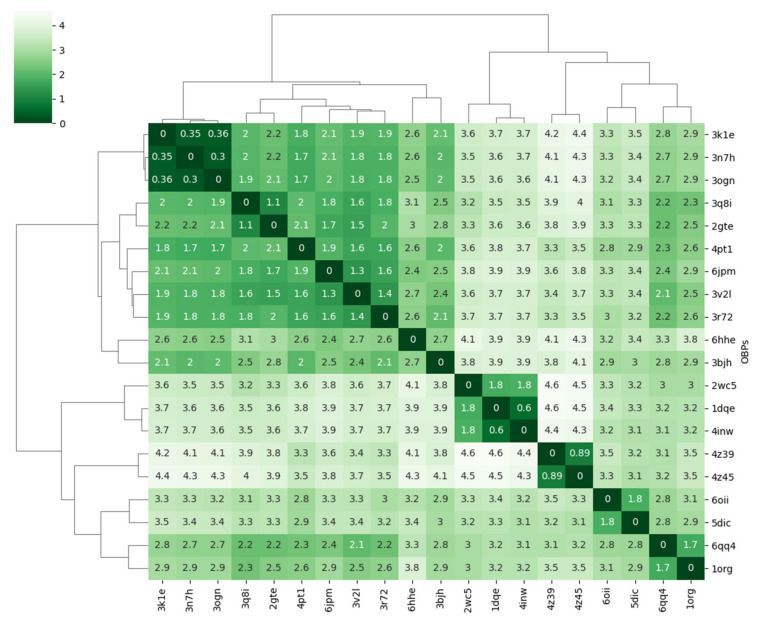
Pairwise similarity matrix of RMSD of equivalent Cα atoms in the MSA. Comparisons of a protein to itself are shown along the diagonal. The values are in Å. The dendrogram shows the hierarchical clustering obtained from the complete linkage algorithm as implemented in the Python package Seaborn.

**Figure 3 biomolecules-12-00282-f003:**
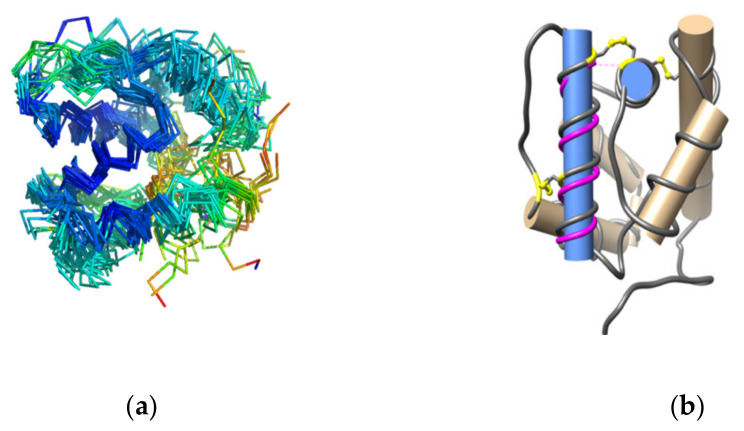
Visualisation of the superposed OBP structures. The helical regions coloured in blue show the least deviation in the positions of backbone Ca atoms (Panel **a**). Geometry of the residues comprising the conserved structural core (residues of the conserved core are depicted in the ribbon coloured in magenta). The two helices of the “conserved” geometry are shown in blue. AgamOBP1 (3n7h) was used in the example (Panel **b**).

**Figure 4 biomolecules-12-00282-f004:**
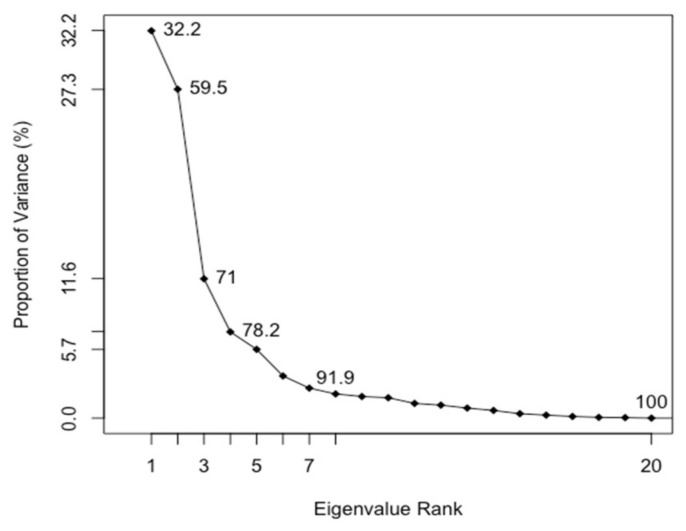
Scree plot. The cumulative variance accounted for by Principal Components. The eigenvalue rank represents the PC number.

**Figure 5 biomolecules-12-00282-f005:**
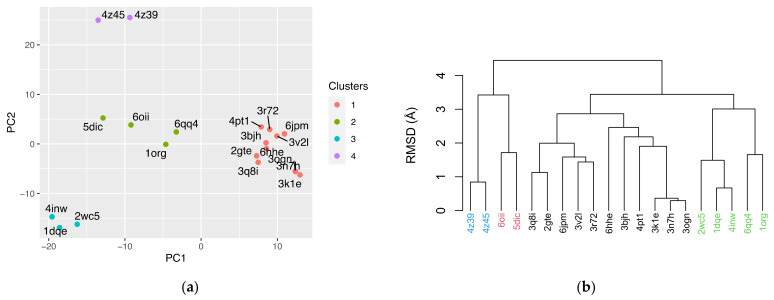
The PC scores in the 20 structures in the dataset, calculated from the deviation in Cα positions. PC1 and PC2 capture 32.2% and 27.3% of the cumulative structural variance, respectively (Panel **a**). Clustering of OBPs based on RMSD similarity (Panel **b**).

**Figure 6 biomolecules-12-00282-f006:**
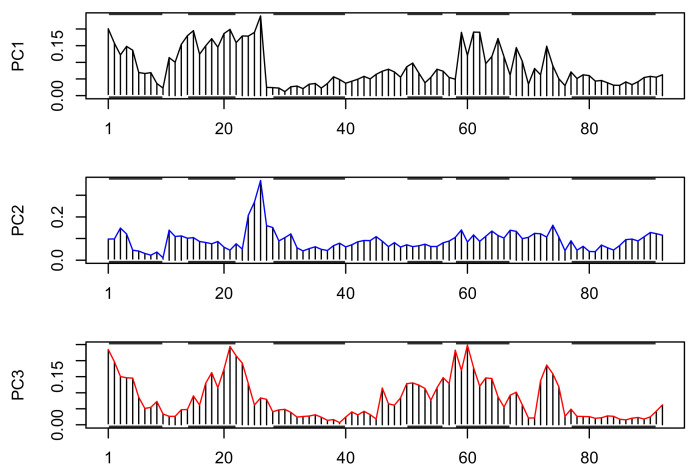
The loadings of the first three PCs. The *X*-axis represents the index of the residues in the multiple sequence alignment in which gaps were removed. The horizontal bars in black represent helical regions.

**Figure 7 biomolecules-12-00282-f007:**
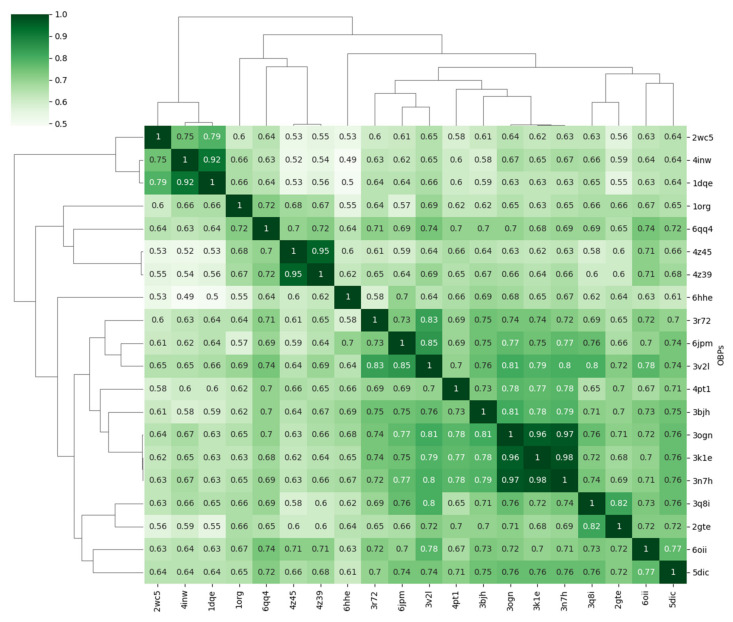
Dynamics-based hierarchically-clustered heatmap. The heatmap is represented such that each structure pairs with all others, including itself. This gives a diagonal with the maximum score of 1.0. The dendrogram was obtained using hierarchical clustering obtained from a complete linkage clustering with Euclidian distances.

**Figure 8 biomolecules-12-00282-f008:**
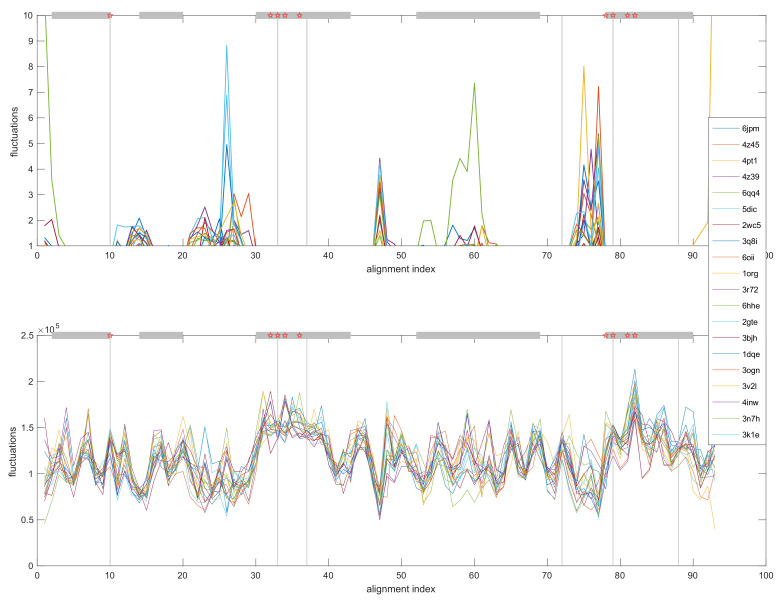
Calculated OBP residue square fluctuations and residue deformation energy profiles. The alignment index refers to columns with 100% occupancy. The horizontal bars in grey show helical regions. * Alignment index positions showing high enrichment in evolutionary coupling interactions.

**Figure 9 biomolecules-12-00282-f009:**
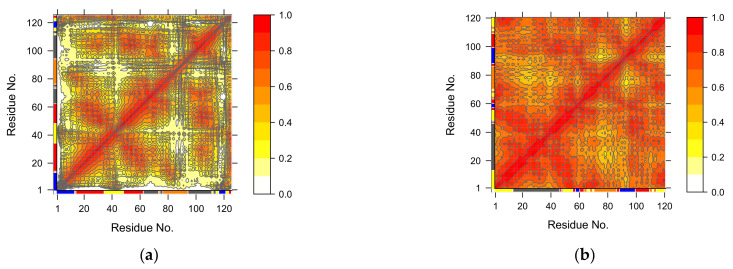
Visualization of the results obtained from the GeoStaS atomic movement similarity matrix (AMSM). AgamOBP1 domain assignment (panel **a**); AaegOBP22 domain assignment (panel **b**). The colour bars at the x and y axes depict the residues assigned to the different domains that were found. The colour scale is the same in both matrices: red indicates strong correlations, and white strong anticorrelations.

**Figure 10 biomolecules-12-00282-f010:**
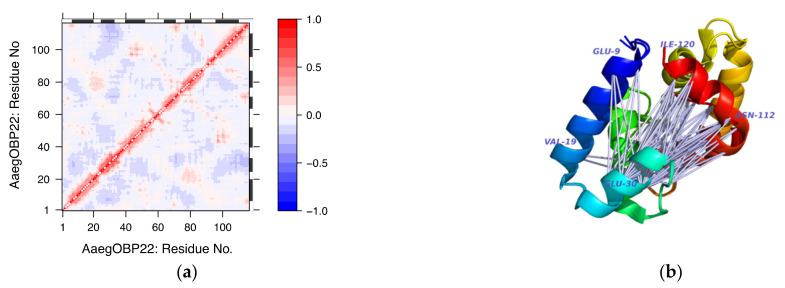
Pairwise residue cross-correlations of AaegOBP22. The cross-correlation matrices were calculated from the complete set of modes. The atomic fluctuations within a system are reflected in the magnitude of all pairwise cross-correlation coefficients for atoms i and j. Correlated fluctuations imply that atoms i and j fluctuate with the same period and phase, whereas anticorrelated fluctuations imply that atoms i and j fluctuate with the same period but opposite phase (Panel **a**). Visualisation of residues showing anticorrelated motions on the superposed structures of the AaegOBP22 in ligand-free and ligand-bound conformations. Correlated motions are not shown (Panel **b**).

**Figure 11 biomolecules-12-00282-f011:**
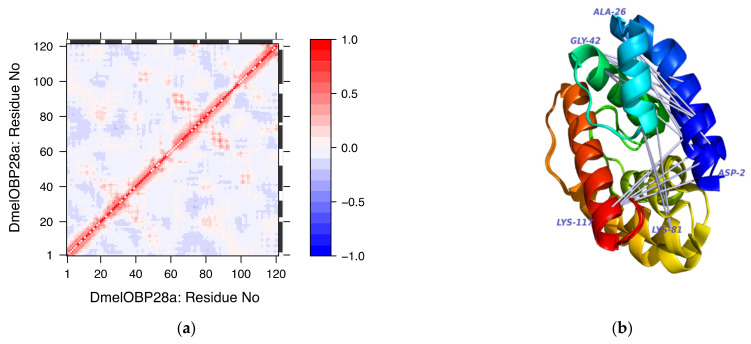
Pairwise residue cross-correlation matrix of DmelOBP28a (Panel **a**). Superposed structures of the DmelOBP28a in ligand-free and ligand-bound conformations. Visualisation of anticorrelated motions shown on the superimposed ligand-free and liganded conformers. Correlated motions are not shown (Panel **b**).

**Table 1 biomolecules-12-00282-t001:** The OBP dataset used in this study.

UniProt	Organism	Abbreviation	PDB ID	Resolution (Å)	No. Residues
Q8T6S0	*Anopheles gambiae*	AgamOBP1	3n7h	1.60	125
Q8T6R7	*Anopheles gambiae*	AgamOBP4	3q8i	2.00	123
Q7Q9J3	*Anopheles gambiae*	AgamOBP20	3v2l	1.80	120
Q6Y2R8	*Aedes aegypti*	AaegOBP1	3k1e	1.85	124
Q1HRL7	*Aedes aegypti*	AaegOBP22	6oii	1.85	120
Q8T6I2	*Culex quinquefasciatus*	CquiOBP1	3ogn	1.30	124
O02372	*Drosophila melanogaster*	DmelOBP76α	2gte	1.40	124
P54195	*Drosophila melanogaster*	DmelOBP28a	6qq4	2.00	121
W8W3V2	*Ceratitis capitata*	CcapOBP22	6hhe	1.52	116
P34174	*Bombyx mori*	BmorPBP1	1dqe	1.80	137
P34170	*Bombyx mori*	BmorGOBP2	2wc5	1.90	141
D0E9M1	*Amyelois transitella*	AtraPBP1	4inw	1.14	140
Q8WRW5	*Apis mellifera*	AmelASP1	3bjh	1.60	117
Q8WRW2	*Apis mellifera*	AmelASP5	3r72	1.15	122
Q3HM32	*Locusta migratoria*	LmigOBP1	4pt1	1.65	129
Q8MTC1	*Leucophaea maderae*	LmadPBP1	1org	1.70	118
A0A0S2E5N6	*Megoura viciae*	MvicOBP3	4z39	1.30	121
A0A0M4AUH6	*Nasonovia ribisnigri*	NribOBP3	4z45	2.02	118
L8B8J6	*Phormia regina*	PregOBP56a	5dic	1.18	115
A0A0R8PDN4	*Chrysopa pallens*	CpalOBP4	6jpm	2.10	119

**Table 2 biomolecules-12-00282-t002:** Summary of the calculated OBP sequence identities.

Minimum	1st Quartile	Median	Mean	3rd Quartile	Maximum
0.090	0.151	0.188	0.244	0.221	1.000

**Table 3 biomolecules-12-00282-t003:** Summary of the calculated OBP sequence identities (values in Å).

Minimum	1st Quartile	Median	Mean	3rd Quartile	Maximum
0.000	2.117	2.876	2.656	3.235	4.446

## Data Availability

Not applicable.

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
