# Peer review of "A Comparative Evaluation of the Structural and Dynamic Properties of Insect Odorant Binding Proteins"

_biomolecules, 2022, doi:10.3390/biom12020282_

Round 1

Reviewer 1 Report

This article addresses structural and dynamic properties of insect odorant binding proteins, the latter being studied by elastic network models. The results are interesting and would benefit a number of researchers around the world working on OBPs from various sources. A few comments - please justify the choice of the OBPs used in the study. These are classical OBPs - the crystal structures are mainly based on having a bound ligand. The PCA plot of Figure 5 should show on the axes labels the information content captured by PC1 and PC2.  Is Figure 8 achieved assuming that there is a ligand bound to the OBP or this is assumed to be free of ligand?  Do the calculations take into account the effect of water molecules and effect of hydrogen bonding on the protein shell and binding pocket?  Interpretation of this in light of the following discussion on the large conformational change induced by ligand binding in DMelOBP28a is extremely interesting - was this the only protein where data was available in ligand-free form?  

There are a few minor typos to be corrected. 

line 164 pairwise

Figure 2 legend Calpha

line 189 these

line 207 modulate

line 360,361 crystallised

Discussion - line 397-398 - it is not clear what is meant - please rephrase.

line 540 Hooke's law

Author Response

Please find attached my detailed response to your comments. My response is in blue typeface.

Thank you for your attention

Reviewer 2 Report

Reviewer #: This is a very well-conceived and written paper. The title: “A comparative evaluation of the structural and dynamic properties of insect odorant-binding proteins“seems to be a relatively wide topic for a title than the subject explained in the publication but the studies explained in it are relevant and support the title. The methods and analysis used are up to date.

Results: With a variety of techniques and analyses the authors have tried to uncover as yet unknown facts on the basis of dynamic structure and properties of OBPs. It’s understandable that the available small datasets on OBPs crystal structures limit overall outcome yet the approaches made for drawing conclusions are commendable.

Discussion: If possible this part can be improved with more relevant available references.

Finally, I think the manuscript contains important issues and interesting approaches which can lead to more progressive future OBPs based strategies. So, I consider this manuscript suitable for publication in Biomolecules.

No grammatical errors. Good figures.

Author Response

Thank you for your feedback.

Please find attached my response to your comments. My response is in blue typeface.

Thank you for your attention

George Tzotzos
